# Influence of the AlSi7Mg0.6 Aluminium Alloy Powder Reuse on the Quality and Mechanical Properties of LPBF Samples

**DOI:** 10.3390/ma15145019

**Published:** 2022-07-19

**Authors:** Irina Smolina, Konrad Gruber, Andrzej Pawlak, Grzegorz Ziółkowski, Emilia Grochowska, Daniela Schob, Karol Kobiela, Robert Roszak, Matthias Ziegenhorn, Tomasz Kurzynowski

**Affiliations:** 1Centre for Advanced Manufacturing Technologies (CAMT-FPC), Faculty of Mechanical Engineering, Wroclaw University of Science and Technology, ul. Łukasiewicza 5, 50-371 Wroclaw, Poland; konrad.gruber@pwr.edu.pl (K.G.); andrzej.p.pawlak@pwr.edu.pl (A.P.); grzegorz.ziolkowski@pwr.edu.pl (G.Z.); emilia.grochowska@pwr.edu.pl (E.G.); karol.kobiela@pwr.edu.pl (K.K.); tomasz.kurzynowski@pwr.edu.pl (T.K.); 2Chair of Engineering Mechanics and Machine Dynamics, Brandenburg University of Technology Cottbus-Senftenberg, Universitätsplatz 1, 01968 Senftenberg, Germany; daniela.schob@b-tu.de (D.S.); robert.roszak@b-tu.de (R.R.); matthias.ziegenhorn@b-tu.de (M.Z.); 3Institute of Applied Mechanics, Poznan University of Technology, Pl. M. Sklodowskiej-Curie 5, 60-965 Poznan, Poland

**Keywords:** additive manufacturing, powder reuse, aluminium alloy, porosity, mechanical properties

## Abstract

Additive manufacturing (AM) is dynamically developing and finding applications in different industries. The quality of input material is a part of the process and of the final product quality. That is why understanding the influence of powder reuse on the properties of bulk specimens is crucial for ensuring the repeatable AM process chain. The presented study investigated the possibility of continuous reuse of AlSi7Mg0.6 powder in the laser powder bed fusion process (LPBF). To date, there is no study of AlSi7Mg0.6 powder reuse in the LPBF process to be found in the literature. This study aims to respond to this gap. The five batches of AlSi7Mg0.6 powder and five bulk LPBF samples series were characterised using different techniques. The following characteristics of powders were analysed: the powder size distribution (PSD), the morphology (scanning electron microscopy—SEM), the flowability (rotating drum analysis), and laser light absorption (spectrophotometry). Bulk samples were characterised for microstructure (SEM), chemical composition (X-ray fluorescence spectrometry—XRF), porosity (computed tomography—CT) and mechanical properties (tensile, hardness). The powder was reused in subsequent processes without adding (recycling/rejuvenation) virgin powder (collective ageing powder reuse strategy). All tested powders (powders P0–P4) and bulk samples (series S0–S3) show repeatable properties, with changes observed within error limits. Samples manufactured within the fifth reuse cycle (series S4) showed some mean value changes of measured characteristics indicating initial degradation. However, these changes also mostly fit within error limits. Therefore, the collective ageing powder reuse strategy is considered to give repeatable LPBF process results and is recommended for the AlSi7Mg0.6 alloy within at least five consecutive LPBF processes.

## 1. Introduction

Additive manufacturing (AM) is dynamically developing and finds applications in industries such as aerospace, automotive, and medicine. The main benefits of AM are the freedom of design, a low level of waste, a decrease in the number of technological operations, the production time, and the supply chain cost in low series production [1,2].

However, certain limits slow down the complete implementation of AM technologies in the industry. For example, a lack of standardization, problems with repeatability, the possibility of quickly moving the process from one machine to another, and insufficient knowledge about powder reuse.

Powder reuse is critically essential for laser powder bed fusion (LPBF). According to ISO/ASTM 52900:2021 [3], LPBF is used to produce objects from powdered materials and uses one or more lasers to fuse or melt the deposited layers of powder particles selectively. Therefore, the possibility of reusing powder in more than one process makes PBF technologies more sustainable and decreases production costs. It is an accepted practice to use the unmelted powder material more than once for the process. Usually, for powder reuse in LPBF, manufacturers use powder recycling/regeneration procedures. However, it is a time- or number-limited usage due to quality and material concerns. Besides that, the authors in the paper [4] explain the difference between recycling (rejuvenation) and reuse. They are limiting the second term to the repetitive use of the same powder, without or with minimal post-processing, such as sieving/screening.

Many authors [5,6,7,8] have been investigating the effect of powder particle size and oxygen content during the LPBF process on the results of its processing. All the researchers agreed that the powder used in consecutive processes must be free of contamination, e.g., oxidation and cross-contamination, which can occur accidentally by mixing metal powders of different chemical compositions [9]. Even spatters formed during the metal powder processing can be treated as contamination, even though the chemical composition remains the same. However, the oxygen content very often increases [10]. These findings push the researchers working on process LPBF development to consider the effect of powder reuse, applied strategies of operating powder between the processes, and the changes of powder characteristics on the result of the AM process.

Various strategies for reusing powders for LPBF can also be found in the literature. For example, the authors of [11] described two of them: (1) continuous refreshing and (2) collective ageing. The first approach states that the powder volume for each build job should be the same. Therefore, all remaining powder is filled with fresh powder after the process, eliminating the powder losses caused by manufactured parts and supports, powder loss due to spatter formation, and machine operation. On the other hand, the collective ageing strategy is about using the once-loaded powder in a machine till the remaining powder amount is insufficient for the next job. Then, the remaining powder volume is filled with a used powder that was sieved and loaded into the main machine tank. Both methods are different in terms of how much powder can be operated in total and how much powder is needed to sieve and mix after every process.

There are many studies on powder reuse in LPBF, including nickel-based alloys [10], iron-based alloys [12], titanium-based alloys [13], and aluminium-based alloys. Recently, published works on aluminium-based alloys focus on the AlSi10Mg [14] and Al-Si-Sc-Zr [15]. The main findings of [15] have pertained to the mechanical properties (ultimate tensile strength (UTS) and elongation at the break of specimens, respectively, built with virgin and reused powder are 565 MPa, 13% and 537 MPa, 11%) and porosity (0.06% for samples made from virgin powder compared to 0.15% for samples made for the reused powder). In addition, the authors of [16] analysed properties of AlSi10Mg alloy aged in air and used for direct-energy deposition (DED); they found out that the increase in oxygen content influences the processability of the powder and the properties of the final specimens.

Aluminium alloys cover about ¼ of the AM powder market by volume [17]. Therefore, the interest in the effective use of these materials and the possibility of reusing the aluminium alloy powders remains at a high level [4]. Understanding the powder degradation mechanism of Al alloy in the LPBF process is essential. The AlSi7Mg0.6 alloy is gaining more and more attention in the AM industry [18,19,20]. However, to date, there is no study of AlSi7Mg0.6 powder reuse in the LPBF process to be found in the literature. This study aims to respond to this gap.

The goal of this study is to evaluate the influence of AlSi7Mg0.6 powder reuse on the material properties of LPBF samples. The powder will be reused in subsequent processes without adding (reuse/rejuvenation step) virgin/fresh powder, using only a double sieving procedure to eliminate the oversized powder particles.

The objectives of this study are:To verify the eventual changes in powder morphology and chemical composition during the continuous reuse;To evaluate the influence of eventual powder degradation on the quality of LPBF bulk specimens.

## 2. Materials and Methods

### 2.1. Materials and Processing

The AlSi7Mg0.6 powder with the chemical composition as listed in Table 1 is used. The powder is supplied by SLM Solutions Group AG (Lübeck, Germany).

Five successive LPBF process cycles are carried out using the SLM 280 2.0 machine (SLM Solutions Group AG, Lübeck, Germany). The LPBF system used in this research is equipped with a 1070 nm fibre laser with a max. power of 700 W and a beam focus diameter between 80 and 115 µm. Each process is held under a protective atmosphere of pure argon (O_2_ level kept below 100 ppm, argon purity class 5.0). In each process, six plates (150 mm × 30 mm × 4 mm) and twelve cylinders (Ø12 mm × 150 mm) are manufactured, as shown in Figure 1a,b. A build volume reduction (100 mm × 100 mm × 150 mm) is used to allow high-volume powder consumption. Parts are manufactured on a 1xxx series aluminium build platform with 98 mm × 98 mm × 20 mm dimensions, heated up to 150 °C and kept at this temperature during the LPBF process. Cylindrical samples are used for microstructure studies, and bar samples are used for tensile testing. To manufacture the samples, previously tested process parameters are used, in line with machine and powder supplier recommendations (SLM Solutions Group AG, Lübeck, Germany). As stated by the supplier, the process parameters should provide sample densities above 99% [21]. The same set of parameters is used for each of the five LPBF processes.

The powder as received from the supplier after initial sieving through 75 μm sieve size is labelled as ‘virgin’ (powder P0), and the powder after each following process cycle is labelled as ‘powder PN’ (powders P1, P2, P3, and P4), where N is a number of powder reuse cycles prior the N process (Table 2). Therefore, the LPBF samples are appropriately named S0, S1 … S4 (in line with the used powder batches P0, P1 … P4). In addition, an oversized powder collected from the sieve was also characterized, labelled as PW (waste powder).

The powder volume is not refilled throughout the experiments. This approach is named “continuous reuse” according to ASTM F3456-22 [22]. Initially, 15 kg of virgin AlSi7Mg0.6 powder was used to manufacture the first batch of samples (S0). After each process, the whole powder volume is removed from the machine and double sieved to eliminate the oversized powder particles. The whole powder volume collected after the N process is sieved in the first step, and then the second sieving is performed only for the overflown powder separated in the first sieving. Finally, the powder that passed through the sieves after the first and second sieving operations is mixed. A powder sieving station PSM 100 (SLM Solutions Group AG, Germany), equipped with a 75 µm flow sieve, is used to sieve the powder.

The total volume of processed powder decreases every build job due to the use for sample manufacturing and rejection after double sieving. The change in powder weight throughout the process cycles is shown in Figure 2. Finally, the height of the last build job has decreased to 120 mm. At the end of the experiment, the weight of powder remaining in use is 11 kg. The part weight to the powder in use weight ratio is about 1:10 and depends on the ‘N’ processing cycle (process 0 to process 4).

Dogbone samples according to ASTM E8/E8M-16 [23] for mechanical testing are machined from the plates produced in each LPBF process (Figure 3). The length of tensile test samples is 100 mm, so the decreased build job height of the last LPBF process does not influence the size of tensile samples.

### 2.2. Powder Characterization

After each process cycle (LPBF + double sieving), the powder is characterised by morphology, flowability, and physicochemical properties. Finally, the results are compared with the properties of waste powder.

#### 2.2.1. Powder Morphology

To measure the particle size distribution (PSD), the HELOS BR R4 + RODOS laser diffraction system, equipped with a VIBRI dispersion unit, is used (Sympatec GmbH, Clausthal-Zellerfeld, Germany). A 68-mbar vacuum and a 2-bar feed pressure are used to disperse particles during testing. A 70% feed rate and 1.5 mm gap width parameters of the VIBRI unit are used to feed the powder. Statistical analysis of the PSD is performed in the PAQXOS 3.1 software (Sympatec GmbH, Clausthal-Zellerfeld, Germany) according to the ISO 13320-1/ASTM B822-17 standards.

The individual powder particles’ morphology and surface condition are characterised by microscopic investigation using SEM EVO MA25 scanning electron microscope (CARL ZEISS, Oberkochen, Germany). The procedure is performed according to ISO 13322-1 standard.

#### 2.2.2. Powder Flowability

A rotating drum (GranuDrum) granular flow analyser is used (Granutools, Awans, Belgium) to characterise powder flowability. The instrument is an automated tester providing the cohesion analysis within the powder flowing in a rotating drum. First, a flowing powder interface position snapshots are analysed. Based on this, the cohesive index is derived. The higher the powder fluctuation during rotation flow, the higher the cohesive index. In addition, the mean avalanche angle is measured during the test. Cohesive index and mean avalanche angles are determined for the drum’s increasing and decreasing rotational speed (hysteresis mode), i.e., 1, 2, 5, 10, 20, 30, 40, 50, and 60 RPM. Thirty flowing powder images are taken at each RPM level with a 1 Hz sampling rate to calculate the cohesive index and mean avalanche angle.

#### 2.2.3. Laser Absorption

Laser absorption assessment is performed with the spectrophotometry method using Exemplar Plus BTC655N-ST laser radiation absorption spectrophotometer (B&W Tek, Newark, DE, USA). The powder sample is placed inside the integrating sphere. Modulated monochromatic light (in a range of 900–1100 nm) is shined at the powder sample. Light reflection is compared to a reference sample with almost 100% reflectivity. Based on the measurement, laser absorption is calculated.

The changes in absorption can be directly translated into the course of the LPBF process and the need to adjust the process parameters (process window) to the current state of the powder.

#### 2.2.4. Chemical Composition

Chemical composition is evaluated using an energy dispersive X-ray fluorescence spectrometer (ED-XRF) SPECTRO XEPOS (METEK, Kleve, Germany). The measurement is done for powder and bulk specimens. The measurement was repeated 3x for one random bulk specimen from each series (S0 … S4) and for each powder type (P0 … P4). Bulk specimens from the XZ plane are prepared as a metallographic sample (polished and ground).

### 2.3. Sample Characterization

#### 2.3.1. Porosity

The quality of the as-built samples is evaluated using a technical computed tomography method. X-ray computed tomography (XCT) enables the reconstruction and evaluation of the external and internal structure of the manufactured samples, which is especially important for samples produced with additive technologies [24,25]. The volumetric models obtained as a result of the XCT reconstruction allow to determine the volume of voids or pores filled with incompletely melted powder (*V_por_*) and the volume of the melted powder (*V_m_*), making it possible to determine the volumetric porosity (*P*) according to the Equation (1) [26]:(1)$$P\;\left[\%\right]=\frac{V_{por}}{V_{m}+V_{por}}\times 100\%$$

The XCT system phoenix v|tome| x m 300/180 (GE Sensing & Inspection Technologies GmbH, Wunstorf, Germany) is used in the study. A micro-focus X-ray tube with a parameter setting (voltage 160 kV and current 120 µA) is used to X-ray the samples. Such parameters of the X-ray tube with a 2K flat-panel digital detector (10-bit) allowed for scanning a set of ROIs (region of interest covering the gauge section) for six samples with a resolution (voxel size) of 19.79 µm. The reconstruction is carried out using dedicated software phoenix datos| x 2.7.2 (GE Sensing & Inspection Technologies GmbH, Wunstorf, Germany) with measurement artefact correction (ring artefact, axis alignment, beam hardening) and noise reduction. Data processing, including data thresholding and porosity detection, is performed using software VG Studio MAX 3.3 (Volume Graphics GmbH, Heidelberg, Germany).

#### 2.3.2. Microstructure Characterization

Metallographic specimens are prepared to analyse the microstructure. The microstructure analysis is carried out on a cross-section parallel to the build direction. The surface of the metallographic specimens is etched with Kroll’s reagent with the following composition:68 cm^3^ H_2_O + 16 cm^3^ HNO_3_ + 16 cm^3^ HF.

The confocal laser scanning microscope (CLSM) LEXT OLS4000 (Olympus, Tokyo, Japan) and the Zeiss SEM EVO MA25 (CARL ZEISS, Oberkochen, Germany) scanning electron microscope (SEM), equipped with an EDS (energy dispersive spectroscopy) analysis system, are used to capture microstructure images.

#### 2.3.3. Mechanical Properties

The static tensile samples are designed following ASTM E8/E8M-16a (room temperatures) [23]. Tensile tests are performed on the HC-25 ZwickRoell servohydraulic testing machine (ZwickRoell GmbH & Co. KG, Ulm, Germany) with a test frame using a ±25 kN load-cell. The tests are carried out with a strain rate of 0.0008 s^−1^ and are continued until the sample brake. Five specimens are used for each series (S0, S1, S2, S3, and S4).

Hardness tests using a Zwick Roel ZHVμ-A hardness tester (Zwick-Roell, Leominster, United Kingdom) are performed. Vickers hardness cross-section profiles are determined at a 2.94 N load (300 g). Five indents are made on each sample used for microstructure evaluation, and a mean value is calculated for each tested sample.

## 3. Results

### 3.1. Powder Characterization

#### 3.1.1. Powder Morphology

In the virgin state (P0), aluminium powder (AlSi7Mg0.6) is characterized by a “potato”-like shape and has some satellites (Figure 4a). Powder samples after each run of the LPBF process (P1–P4, Figure 4b–e) are characterized by a similar morphology and demonstrate a lack of changes due to processing. However, the waste powder (PW) is significantly different—it has a different surface morphology, particles are 2–3× bigger than virgin powder particles, and lacks satellites.

The obtained particle size distributions (PSD) for each evaluated powder are comparable. There is no significant difference between each consecutive sieving cycle (P0 to P4). The curves for powders P0–P4 are plotted simultaneously one on another, both for distribution density q3 (Figure 5a) and the cumulative density Q3 (Figure 5b), showing almost identical plots. The SEM observations and PSD analysis show the presence of satellite particles throughout the powder states with no variations. The values of x_10,3_, x_50,3_, and x_90,3_ characteristic particles size parameters are presented in Table 3. In the case of the PW probe, PSD curves are clearly visible and are moved to the right side, which means that the powder sample consists of much larger diameter particles. PW x_50,3_ is 2.5-times higher than P0–P4 and x_90,3_ is 3.5-times higher than P0–P4.

The PS distribution of PW is not a normal distribution, unlike the PSD of P0–P4. This suggests that the waste powder is not homogeneous in its volume. Based on this observation, the waste powder (PW) is a mixture of a non-remelted powder, and a re-melted spatter powder.

#### 3.1.2. Powder Flow Properties

The measured flow properties of the powders in each of the P(N) states behave in a very similar way. The measured flow properties of the powders in each of the P(N) states behave in a very similar way. All powder samples are characterized by a relatively high cohesive index and a tendency to increase a cohesive index with an increasing drum speed. According to [27], it is influenced by particle size and shape. An increasing drum speed increases bonding between the particles and influences static (angle of repose) and dynamic (cohesive index and flow) behaviour.

Based on the plots in Figure 6, there is no difference between P0 and P4 powders. Therefore, only the curve for the PW sample can be differentiated from the reused powders. The cohesive index for the waste powder slightly increases up to 10 RPM, and with an increase in the rotational speed, the cohesive index remains constant (Figure 6a). When analysing the trend of the avalanche angle for the PW sample, again, 10 RPM is a threshold where the behaviour changes, and below 10 RPM, the avalanche angle is constant. With a higher rotation speed, the measured avalanche angle is comparable to the sieved P0–P4 powders used in the consecutive processes (Figure 6b).

#### 3.1.3. Physicochemical Properties

The absorption measured in the range of 1020–1100 nm of the wavelength is comparable, and in the case of P0–P4 powders is between 57 and 62%. Waste powder PW exhibits a 25% higher absorption and reaches a value of ~78% (Figure 7a). A slight difference in the sample after the first manufacturing process (P1) can be distinguished by zooming in the plot to a narrower absorption scale (Figure 7b). It can be seen that the registered signal in the entire wavelength range is about 5% higher than the rest of the analyzed samples. A polynomial curve fitting of the laser absorption measurement with a 95% confidence interval confirms the observed difference between P1 and P0, P2, P3, and P4 samples.

#### 3.1.4. Chemical and Phase Composition

In order to determine the influence of the powder degradation on the functional properties of the samples obtained by the LPBF method, the chemical composition analysis using the XRF method was performed. Table 4 and Figure 8 show the analysis results for the bulk samples (S0 … S4) and the powders (P0 … P4).

The chemical composition remains unchanged. The minor discrepancies are within the error limits and the XRF method accuracy. In the case of aluminium alloys, the evaporation of some elements (for example, Mg or Zn) is one of the critical aspects of LPBF processing [28]. In the present study, it can be noticed that the magnesium content in the alloy did not change significantly in the case of both the powder and solid samples.

### 3.2. Sample Characterization

#### 3.2.1. Porosity

The XCT analyses aimed to assess the internal structure of the samples manufactured from successive iterations of the reused powder and to check the impact of powder degradation on the defects’ formation. For this purpose, six samples from each series were scanned, as presented in Figure 9.

The porosity analysis was performed in the same method for each sample for a region of interest (ROI) 25 mm high (Figure 9a), corresponding to the gauge section of the tensile sample (Figure 3). The results of the porosity analyses are presented in the form of graphs of the mean values of (a) porosity, (b) maximum pore diameter, and (c) maximum pore volume, taking into account the standard deviation of the results (Figure 10).

Based on the obtained results, there is no noticeable trend of changes in the porosity of the samples concerning subsequent iterations of the input material processing. The exception is series 4four for which the recorded porosity values are the lowest. However, the results are evenly distributed for all series concerning the mean value of the maximum diameter size and pore volume.

The diameters of the registered pores and their shape (sphericity) were evaluated for three samples showing the highest porosity from each series. In Figure 11a, box plots of pore diameters are presented. In Figure 11b, box plots of pore sphericities are presented. Pore sphericity is defined as the aerial ratio of the sphere to the pore where the sphere outlines the pore. The closer the value to one, the higher the pore sphericity is [29].

The results of the data from the individual series and between all the processes coincide with each other. There is a slight difference in the mean pore sizes for all S4 samples compared to the rest of the series (S0–S3). There is also a slight decrease in mean sphericity. Nevertheless, pore diameter and sphericity show significant deviation, here expressed as 1.5 times the interquartile range (1.5IQR). Additionally, each series and sample show outliners. A low number of pores in each series show larger diameters than 1.5IQR. A low number of pores show also smaller and larger sphericities than 1.5IQR.

The total number of pores recorded in the ROI volume distinguishes the S0–S4 series and the S4 series. The pore number is about 30% smaller for the S4 series than for the S0–S3 series. A comparison of the pore morphology and pore count is presented in Figure 12.

The samples’ pore distribution homogeneity was compared for the XZ, YZ, and ZX (Figure 13a–e) and samples S0-1 and S4-1. The porosity analysis was performed for sequential ROIs with heights equal to 0.25 mm [25]. The most significant changes in porosity were noted for the XY plane. It results from the presence of subsurface pores, which are a defect caused by the selected LPBF boundary process parameters and strategy. Hence, the porosity in this plane has the highest values, even up to 1.35% (Figure 13f). It is worth noting that this phenomenon does not occur in the other planes of the analysis due to the machining and ROI selection. Hence, the global values do not exceed 0.22%, as shown in Figure 13g,h.

#### 3.2.2. Microstructure

The microstructure of the samples is typical for the additively manufactured aluminium alloys and is characterized by a fine, columnar–dendritic structure. The CLSM (Figure 14) and SEM (Figure 15) microscopic images do not differ between each series. Figure 14a–e represent the XZ plane of specimens, and Figure 14f shows the XY plane of an S0 specimen to show the characteristic fusion lines and material texture in planes parallel (XZ) and perpendicular (XY) to the LPBF build direction (BD).

The microstructure of AlSi7Mg0.6 is typical for hypoeutectic alloy [30] (Figure 15a–e). It consists of the α-Al phase (grey background on SEM microphotographs) and a network of inter-dendritic regions rich in Si (light grey colour).

On SEM micrographs, the dendrite arm spacing was measured (Figure 15f) to assure no difference and no influence of powder reuse on the microstructure features. The results of measurements are presented in Figure 15f, and there are minor differences within samples, but all of the measurement differences fall within the error limits. Therefore, it can be concluded that there are no differences, especially considering that such a measurement will be sensitive to the collection site and the orientation of the specimen cross-section to the examined dendritic structure.

### 3.3. Mechanical Properties

Performed static tensile tests showed no visible powder degradation trend due to its reuse. All series show a high UTS of 390–400 MPa. The highest mean values were obtained for samples from series S0, S1, and S2, with narrow confidence intervals (Figure 16a) and UTSs above 395 MPa. The series S3 and S4 have lower mean values (below 395 MPa), but the confidence intervals reach the mean value of the rest of the analysed series.

The strain at break values shows high consistency and each confidence interval overlaps. Each series show 6–7% of strain at break (Figure 16b).

The hardness of the samples (mean for all samples of 118 ± 3 HV0.3) is comparable with the value from a material datasheet (112 ± 3 HV10) [21]. Furthermore, each confidence interval overlaps. The summary for the hardness measurements for each series of samples (S0–S4) is presented in Figure 16c.

The obtained mechanical properties are in line with properties of AlSi7Mg0.6 alloy processed by LPBF found in the literature (Table 5). Despite the slight differences in UTS, hardness, and strain at break, the typical correlation is maintained. A lower hardness results in a higher elongation and a lower UTS. The phenomena are related to the parameters of the LPBF process and resulting solidification. Faster cooling creates finer, less ductile microstructures, thus producing higher UTS [31].

## 4. Discussion

The hypothesis based on the literature data about other powder materials [32,33,34] was that AlSi7Mg0.6 powder would degrade with each cycle: its surface would be oxidised and PSD would increase toward bigger particles [35]. Therefore, it was expected that the laser absorption would change due to the surface oxidation and PSD, impacting the process conditions and the final sample properties. Furthermore, the literature claimed that such surface oxidation is typical for highly reactive materials such as titanium [6], nickel [10,36], and aluminium alloys [37]. However, the results shown above demonstrate that the analysed AlSi7Mg0.6 powder is highly stable in terms of the laser absorption level during five consecutive LPBF processes.

The minimal absorption increase for the P1 powder sample can be connected to multiple reflections. According to [38], it can appear when the beam is reflected away by the powder bed more than one time. Authors of [38] mention that the reason for multiple reflections is the grain size and shape. Both the particles smaller than the beam’s diameter and non-spherical particles can cause the multiple reflections and therefore increased the absorption. Therefore, the measurement results could be influenced by the powder layer composition during the measurement or the place of powder sampling.

Nevertheless, if the ranges of the laser absorption values are compared and not the values of the averaged polynomial curve fitting (Figure 7), the difference in the series P1 is less significant. Besides, it should be mentioned that the surface structure significantly impacts the absorption level [39]. Therefore, a significantly higher absorption of the waste powder (PW) confirms a surface structure and PSD influence on the laser absorption level.

The other characteristic is powder morphology. Two distinctive features of the powder set it apart from the virgin powders reported in the powder reuse literature. Therefore, it should be considered:(1)Small powder particles (in the form of satellites and loose particles) are usually found in virgin powders [10,15]. Such powders during reuse are losing small particles. Therefore, the changing PSD translates into the change of powder flow or laser absorption [10]. The powder analysed in this work does not have many small powder particles. Moreover, as mentioned in Section 2.1 (materials and processing), virgin powder was pre-sieved before the first P0 process. That is why there is no significant difference in PSD analysis. The sieving procedure between processes successfully separate agglomerations and partial melted particles, which can impact the process.(2)The shape of particles. The analysed powder has elongated, potato-like shape particles. However, it shows an acceptable level of flowability and processability. The multiple processing of AlSi7Mg0.6 powder in LPBF does not change its flow properties compared to other materials such as titanium [40] or Inconel 718 [10] powder. According to the [41], the flowability can be even improved between 6 and 15 cycles of reuse.

In terms of chemical composition, there were two expectations or hypotheses. The first one is about a general change of chemical composition due to multiple powder reuse. The main difference considered is the change of zinc. However, this change is minor and does not affect the properties of material.

The second one was about magnesium evaporation [28]. As a result of five consequent processes, magnesium’s evaporation was not detected in powder or bulk samples. According to [42], the small addition (low content) of magnesium (up to 1.5%) positively influences the microstructure and processability of Al-Si alloys. It should be emphasised that magnesium in low-magnesium Al-Si alloys tends to condense and be studded at the cell boundaries, especially at the nodes of cell boundaries [42]. Therefore, the results described above are well-aligned with those in the literature.

However, magnesium evaporation is the main problem in AM of high-magnesium Al-Mg alloys, but is not confirmed by the literature for the Al-Si alloys except for one publication [43], where authors describe the evaporation of magnesium and zinc for different aluminium alloys. Process parameters are possible reasons for the difference between the results published in [43] and the presented research. High-power LPBF processing should contribute to this effect. An interconnection between magnesium content, densification, and applied energy density affects the influence of process parameters on magnesium evaporation. In the case of magnesium content <2.0%, there is no need to use high-energy density to densify samples [42]; therefore, the risk of magnesium evaporation is lower.

As for the powder samples, the analysed AlSi7Mg0.6 bulk specimens (S0–S4) do not show significant proofs of powder degradation. All bulk specimens (S0–S4) show comparable and repeatable microstructures (similar texture and dendrite arm spacing), which are typical for AM-processed hypoeutectic aluminium alloys.

A similar consistence of results is found within mechanical properties and hardness. The difference between each series (S0–S4) is lower than the standard deviation of the results. Even if the error is neglected, the determined mean values are within 5 MPa for the UTS, 0.5 p.p. for the break at strain, and 5 HV0.3 for the hardness.

In the paper [44], the influence of powder reuse on the mechanical properties of AlSi10Mg alloy within eight consecutive LPBF processes without rejuvenation is presented. The study shows that significant (visible) changes can appeared after the 5–6 LPBF processes. However, in the discussed paper, error limits for each series are not presented. If the changes can be fitted into the error limits as in this work, therefore AlSi10Mg powder degradation shown in [44] could be minor.

On the contrary, if results from [44] are considered, series S4 may be the critical point, after which some changes could appear. Even if all powder samples (P0–P4) and bulk specimens (S0–S4) show repeatable properties with changes within error limits, certain signs could indicate some initial degradation. A small decrease in dendrite arm space can be observed, a small change in chemical composition and a higher mean hardness. All the above-mentioned minor changes can be translated into the change of detected bulk samples’ porosity distribution. The pore count is approximately 30% lower for the S4 series than for the S0–S3 series.

The authors of [4] showed in their work the difference between two different powder reuse strategies. The strategy used in the presented work (continuous reuse/single batch) has its limit regarding powder availability for producing the subsequent samples. At a certain moment, it is impossible to process samples with the same geometry due to the lack of powder. Therefore, the number of cycles is limited by the quantity of powder without rejuvenation.

The approach with frequent refreshing from one point is more similar to real production conditions. However, at the same time, the powder degradation during the following cycles is levelled/slowed down by the constant addition of virgin powder.

In the presented study, an attempt was made to maintain identical LPBF processing conditions at each stage of collective ageing powder reuse. Each of the LPBF processes was carried out using the same parameters, samples with a constant cross-section were fabricated, and the conditions of the LPBF process were strictly controlled (platform temperature, pressure in the chamber, gas flow speed, oxygen level, laser beam power variation, etc.). The variability of the parameters recorded during LPBF processes did not exceed 5%. In addition, the powder after each step was screened twice to ensure the adequate separation of oversized particles. The experiment was stopped after five cycles as the amount of powder that remained in circulation was insufficient to allow the fabrication of full height tensile samples. During the process, approximately 30% of the initial amount of powder (15 kg) was used to fabricate samples (4.8 kg) and 0.3 kg (2%) was screened as waste. Given the above, the analysed AlSi7Mg0.6 powder showed a high stability during reuse in the LPBF process. After the five consequent processes run without adding fresh (virgin) powder, it is demonstrated that most properties of both powder and bulk samples remain unchanged.

Therefore, in the case of the AlSi7Mg0.6 alloy, these are conditions for which the collective ageing powder reuse strategies should be safe and repeatable. The present study’s future scope is to analyse the limit of safety of AlSi7Mg0.6 powder reuse. So far, the five cycles of continuous reuse do not influence the quality of produced samples. It will be essential to design the experiment looking for those limits and create the roadmap for the first signs of powder degradation.

## 5. Conclusions

The presented study investigated the possibility of continuous reuse (collective ageing strategy) of AlSi7Mg0.6 powder in the laser powder bed fusion process. The five batches of AlSi7Mg0.6 powder (P0–P4) and five bulk LPBF samples (S0–S4) series were characterised for powder morphology, chemical composition, porosity, and microstructure. In addition, the mechanical properties of the LPBF AlSi7Mg0.6 specimens fabricated with reused powder were investigated to ensure the comparable properties of each reuse cycle. Based on the presented results, the following conclusions can be drawn:The average particle size, morphology, and chemical composition of the virgin and continuously reused AlSi7Mg0.6 powders are comparable. The main outliner is waste powder, screened during double-sieving, which differs in each property from the virgin and continuously reused power.Mechanical properties of the LPBF AlSi7Mg0.6 samples manufactured using continuous reused powder are similar to the LPBF AlSi7Mg0.6 alloy manufactured samples using virgin powder. It confirms that the approach of continuously reused powder can be reasonably used in the LPBF process without a negative effect on the quality of the final product.The collective ageing powder reuse strategy is considered to give repeatable LPBF process results and is recommended for the AlSi7Mg0.6 alloy within at least five consecutive LPBF processes.The presented findings should be only considered when: LPBF process parameters are strictly controlled; the powder is double-sieved in each process; the virgin powder shows a similar morphology to the powder used in this study—it is free from small powder particles and is pre-sieved before use.Samples manufactured within the fifth reuse cycle (series P4, S4) showed signs indicating initial degradation. These changes, however, mostly fit within error limits. Further studies should be looking at the high-cycle reuse of AlSi7Mg0.6 alloy in LPBF to set the reuse limit and create the roadmap for the first signs of powder degradation.

## Figures and Tables

**Figure 1 materials-15-05019-f001:**
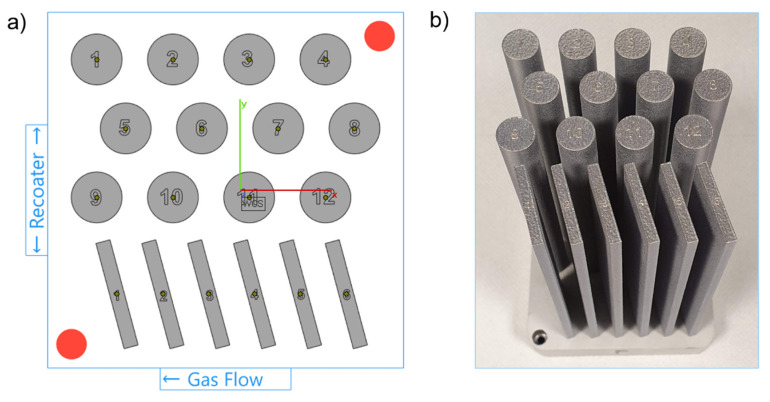
(**a**) LPBF process build job layout; (**b**) An exemplar build plate with samples manufactured.

**Figure 2 materials-15-05019-f002:**
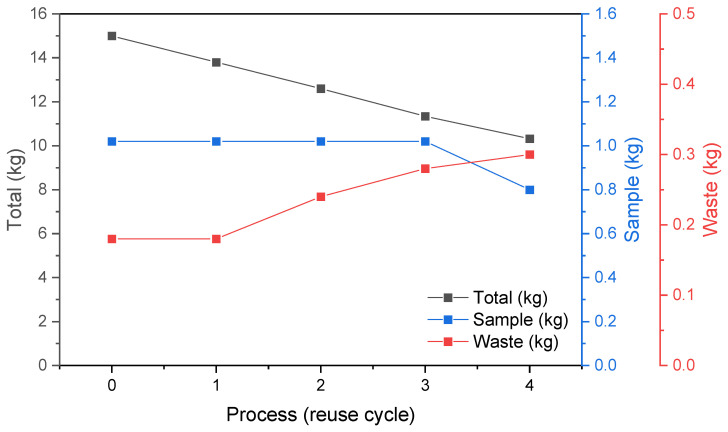
Summary of the total powder weight used in the study, including the weight of obtained samples and the weight of waste powder after double sieving.

**Figure 3 materials-15-05019-f003:**
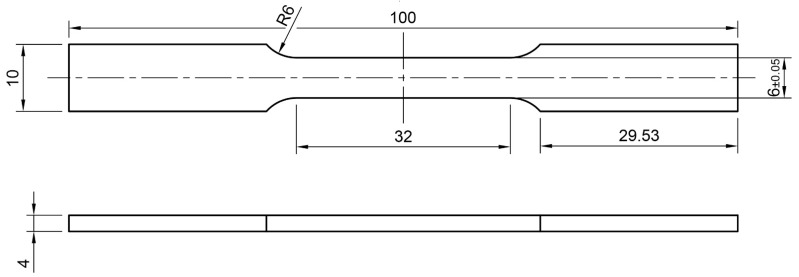
Tensile specimens’ geometry according to ASTM E8/E8M-16 (all dimensions are in mm).

**Figure 4 materials-15-05019-f004:**
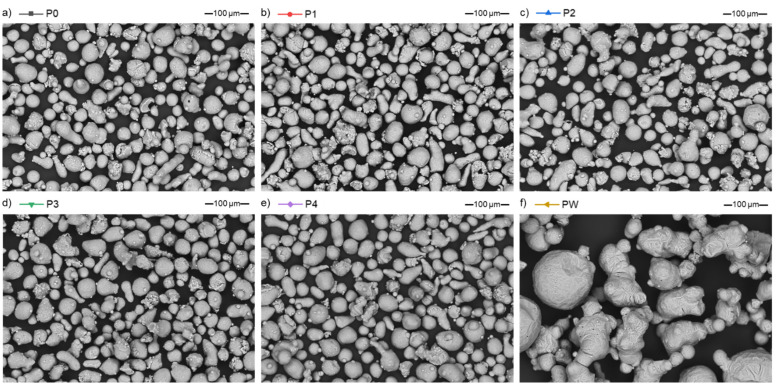
Microscopic images of the AlSi7Mg0.6 powders in various processing states, SEM/BSE: (**a**) virgin (P0); (**b**) after 1st LPBF process (P1); (**c**) after 2nd LPBF processes (P2); (**d**) after 3rd LPBF processes (P3); (**e**) after 4th LPBF processes (P4); and (**f**) waste (PW).

**Figure 5 materials-15-05019-f005:**
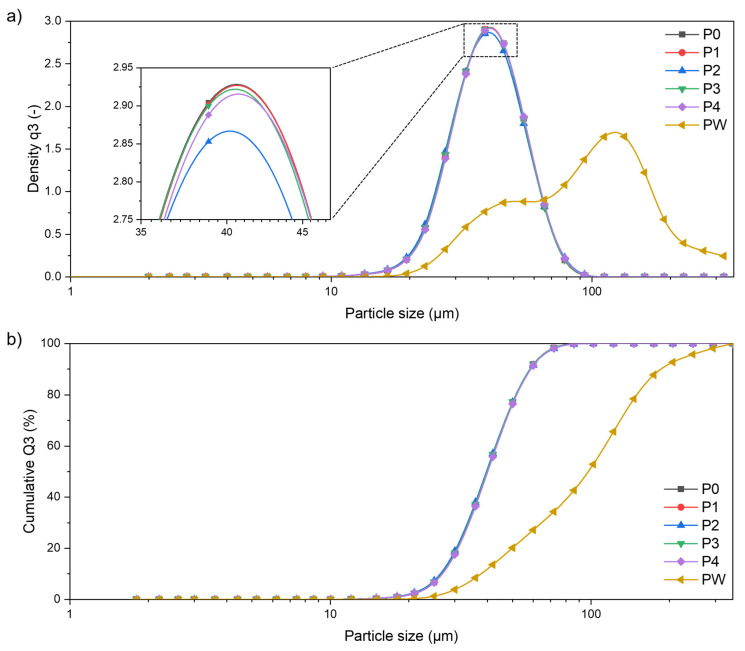
Particle size distribution of AlSi7Mg0.6 for each of the powder states: (**a**) distribution density q3; (**b**) cumulative density Q3.

**Figure 6 materials-15-05019-f006:**
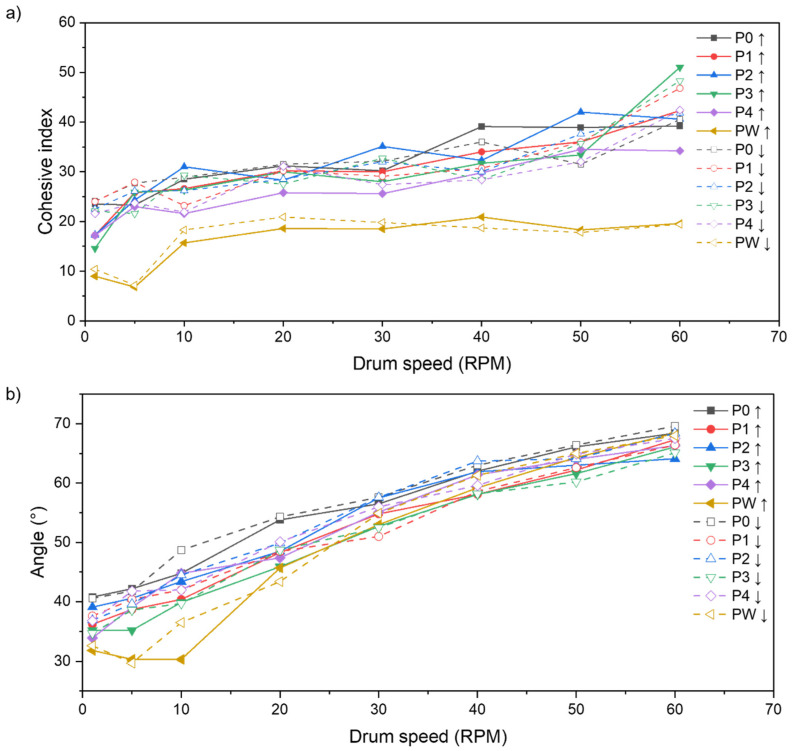
(**a**) The cohesion index for the AlSi7Mg0.6 powder in different processing states regarding the rotational speed measuring drum; (**b**) Mean avalanche angle values for the AlSi7Mg0.6 powders in various states with regards to the rotational speed of the measuring drum.

**Figure 7 materials-15-05019-f007:**
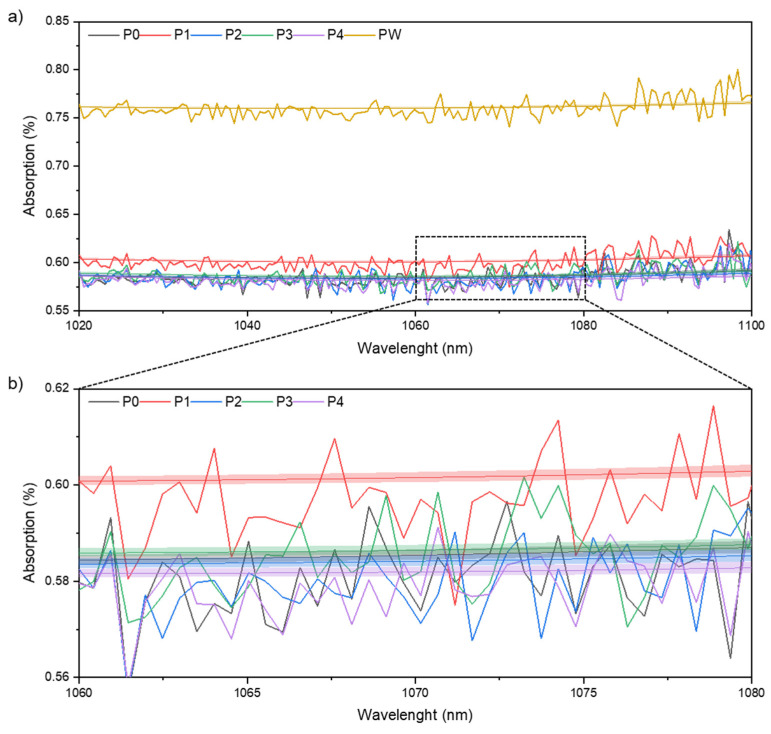
Laser light absorption for wavelengths from 1020 to 1110 nm of AlSi7Mg0.6 powder and polynomial curve fitting with 95% confidence interval (**a**) P0, P1, P2, P3, and P4 state and PW absorption for wavelength from 1020 to 1110 nm for (**b**) magnification for the wavelength from 1060 to 1080 nm and for P0, P1, P2, P3, and P4 states.

**Figure 8 materials-15-05019-f008:**
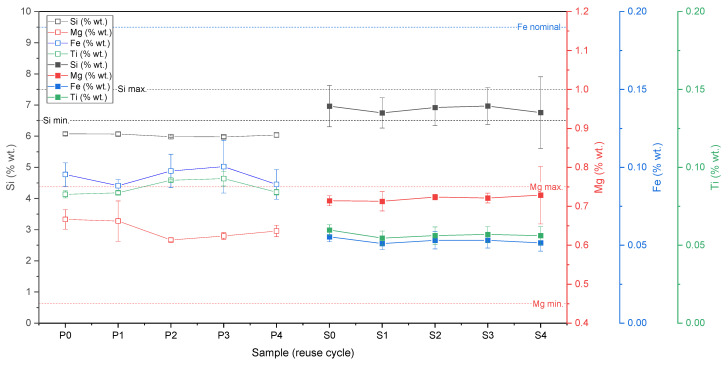
Chemical composition of powder (in different state—P0 … P4) and bulk samples (S0 … S4) from AlSi7Mg0.6 in wt.% measured by XRF method.

**Figure 9 materials-15-05019-f009:**
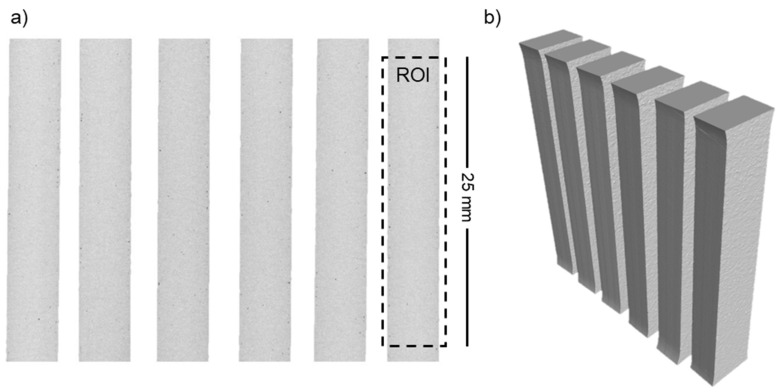
Two-dimensional cross-section through the reconstructed sample (**a**) and a three-dimensional view of the obtained models (**b**). The reconstruction looks similar for all series. Presented reconstruction refers to series S0.

**Figure 10 materials-15-05019-f010:**
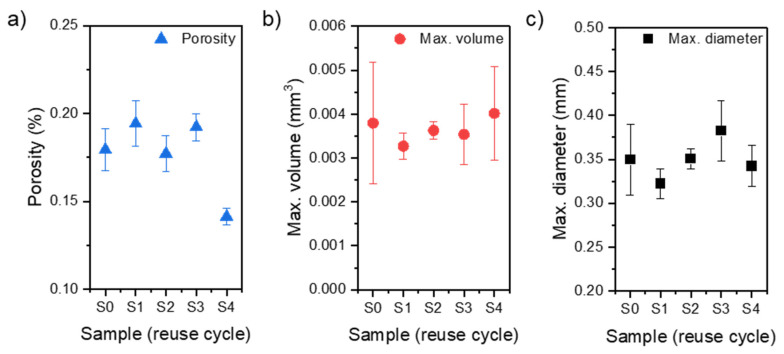
Analysis of the porosity of the measuring part of the samples (error—95% confidence interval): (**a**) a graph of measured porosity (XCT), (**b**) a graph of the maximum pore diameter, and (**c**) a graph of the maximum pore volume.

**Figure 11 materials-15-05019-f011:**
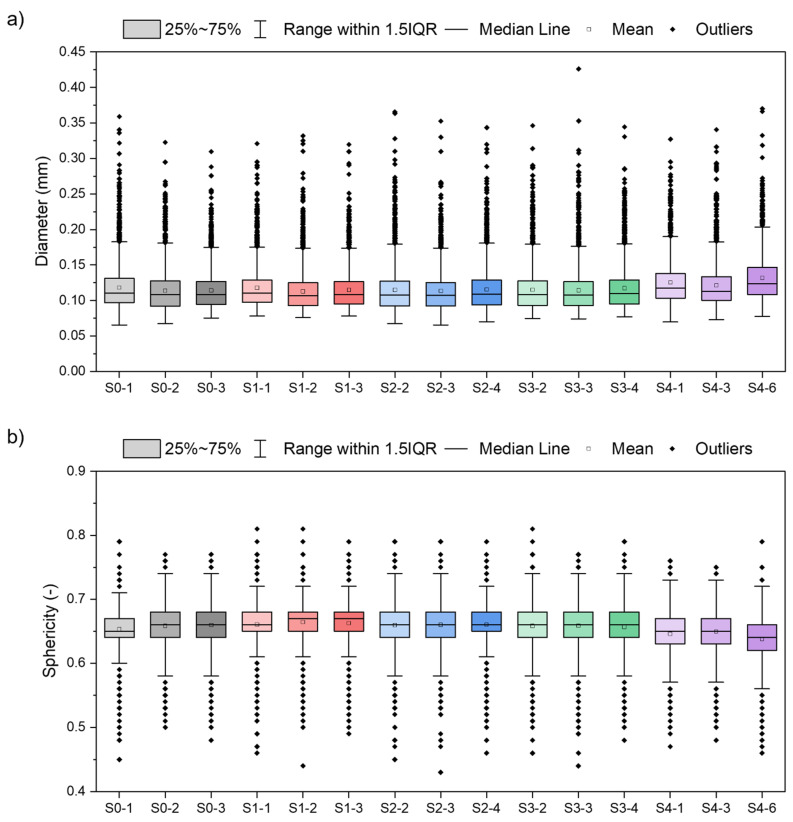
Box plots of pore diameter (**a**) and pore sphericity (**b**) based on CT data for three of the most porous samples of each series: S0 to S4; outliners qualified using the 1.5IQR method (IQR—interquartile range).

**Figure 12 materials-15-05019-f012:**
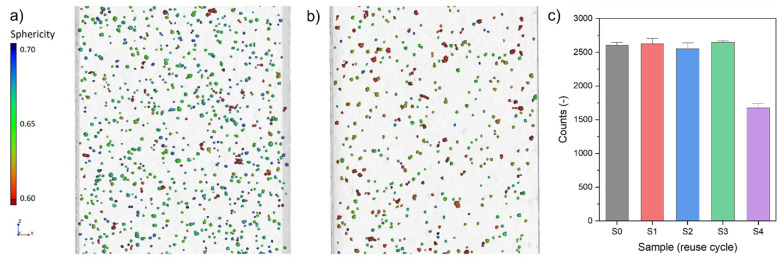
Pore reconstruction recorded for samples (**a**) S1-1 and (**b**) S4-6; (**c**) Pore count in ROI for each respective series (S0–S4)—histogram based on samples analysed in Figure 11 (error—95% confidence interval).

**Figure 13 materials-15-05019-f013:**
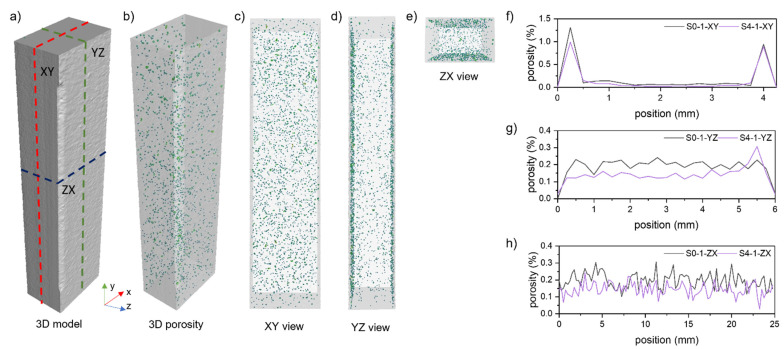
Pore distribution in the sample S1-1 (**b**), visible in planes (**a**), XY (**c**), YZ (**d**), ZX (**e**), and graphs of porosity in the analysed planes (**f**–**h**).

**Figure 14 materials-15-05019-f014:**
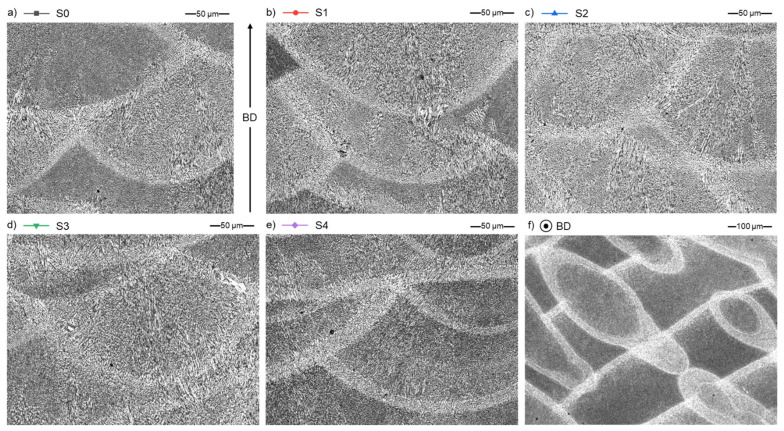
General microstructure of AlSi7Mg0.6 samples cross-section parallel to build direction (BD) for S0–S4 and perpendicular for S0, CLSM; (**a**) S0; (**b**) S1; (**c**) S2; (**d**) S3; (**e**) S4; and (**f**) S0.

**Figure 15 materials-15-05019-f015:**
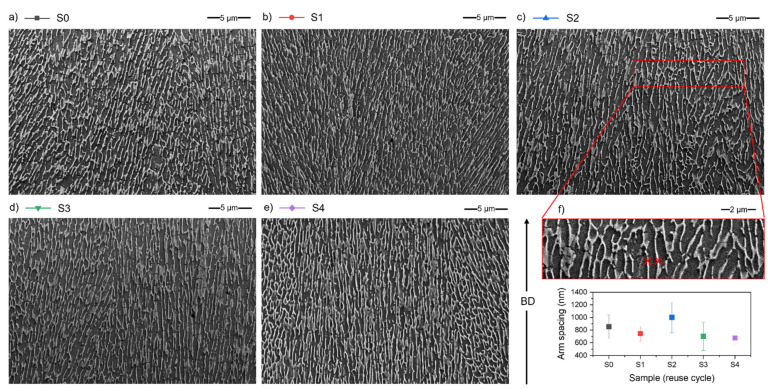
Microstructure of AlSi7Mg0.6 samples, SEM/BSE; (**a**) S0; (**b**) S1; (**c**) S2; (**d**) S3; (**e**) S4; and (**f**) Dendrite arm spacing measurement example and a graph of dendrite arm spacing—avg. from 6 random measurements for each sample from series S0 to S4.

**Figure 16 materials-15-05019-f016:**
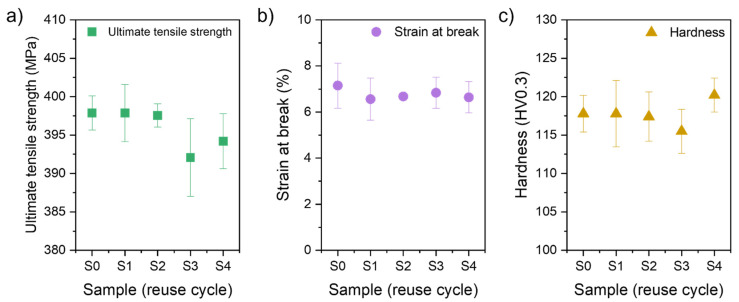
Mechanical properties of LPBF AlSi7Mg0.6 samples S0–S4 (error—95% confidence interval): (**a**) Ultimate tensile strength, (**b**) Strain at break, and (**c**) Hardness (HV0.3).

**Table 1 materials-15-05019-t001:** Chemical composition of the AlSi7Mg0.6 powder used in this study.

Standard	Al	Si	Mg	Ti	Fe	Cu	Mn	Zn	Other Total
EN AC42200 acc. to EN-1706, wt.%	Bal.	6.5–7.5	0.45–0.70	-	0.15–0.19	0.03–0.05	0.1	max 0.07	0.10
SLM Solutions, wt.%	Bal.	6.5–7.5	0.45–0.70	0.25	0.19	0.05	-	max 0.07	0.10

**Table 2 materials-15-05019-t002:** The list of powder samples used in the research.

Powder Sample	Description
P0	Initial batch of virgin powder. P0 powder is dried and sieved before use.
P1, P2, P3, P4	Powder after 1, 2, 3, and 4 LPBF process cycles and double sieving.
PW	Waste powder. The powder that stayed on the sieves after double sieving.

**Table 3 materials-15-05019-t003:** Particle size distribution of AlSi7Mg0.6—volume-weighted characteristic values obtained in the laser diffraction measurements for each of the powder states.

Parameter	P0 (µm)	P1 (µm)	P2 (µm)	P3 (µm)	P4 (µm)	PW (µm)
x_10,3_	26.50	26.50	26.14	26.40	26.54	37.78
x_50,3_	40.02	40.07	39.76	39.94	40.21	97.50
x_90,3_	58.75	58.85	58.97	58.73	59.08	188.34

**Table 4 materials-15-05019-t004:** Chemical composition of powder (in different states) and samples manufactured from each powder in wt.%.

Composition	Al	Si	Mg	Fe	Ti	Cu	Zn	Other Each	Other Total
AlSi7Mg0.6—SLM Solutions—materials datasheet	Bal.	6.50–7.50	0.45–0.70	0.19	0.25	0.05	0.07	0.03	0.10
**Powder specimens**									
P0	Bal.	6.13	0.64	0.09	0.08	0.001	0.010	-	-
P1		6.13	0.72	0.09	0.085	0.001	0.007	-	-
P2		5.96	0.61	0.11	0.092	0.001	0.001	-	-
P3		6.04	0.62	0.12	0.098	0.001	0.001	-	-
P4		6.11	0.65	0.10	0.082	0.001	0.001	-	-
**LPBF specimens**									
S0	Bal.	6.20	0.729	0.052	0.056	0.0006	0.004	-	-
S1		6.188	0.715	0.049	0.052	0.0006	0.004	-	-
S2		6.25	0.727	0.049	0.053	0.0007	0.004	-	-
S3		6.288	0.735	0.052	0.056	0.0007	0.004	-	-
S4		6.220	0.729	0.051	0.057	0.0006	0.004	-	

**Table 5 materials-15-05019-t005:** Mechanical properties of LPBF AlSi7Mg0.6 specimens produced in XZ direction (comparison of different sources).

Property	This Research *	SLM Solutions [21]	Pereira et al. [30]
UTS, MPa	398 ± 2	375 ± 17	435 ± 18
Strain at break, %	7 ± 1	8 ± 2	3 ± 1
Hardness, HV	119 ± 2.5	112 ± 3	136.4 ± 2.5

* Presented values concerning series 0 (virgin powder).

## Data Availability

Not applicable.

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
