# Peer review of "Influence of the AlSi7Mg0.6 Aluminium Alloy Powder Reuse on the Quality and Mechanical Properties of LPBF Samples"

_materials, 2022, doi:10.3390/ma15145019_

Round 1
Reviewer 1 Report
1. What is dendrite width?
2. Authors mentioned that "series 4 (S4) is characterized by a decrease in general porosity level, but at the same time the average size of pores is increased." Write a few more reasons in the abstract.
3. Write the objective before section 2.
4. Mention ASTM E8 with reference.
5. provide a reference for equation 1.
6. Provide experimental setup for Zwick Roel ZHVμ-A hardness tester.
7. Write future scope.
8. Add more recently published literature.
Author Response
We would like to refer to all of the Reviewer’s comments and the changes made to the manuscript; therefore, all significant changes are highlighted in red colour in the revised manuscript. Please find below our answers to all of the Reviewer’s specific comments in the attachment.

Reviewer 2 Report
The authors reported the manuscript entitled "Influence of the AlSi7Mg0.6 aluminium alloy powder reuse on the quality and mechanical properties of LPBF samples" very well. Powder reuse is critically important for laser powder bed fusion including several alloys. This article described evaluating the influence of AlSi7Mg0.6 powder reuse on the material properties of LPBF samples. The article is well written but some points may be cleared before publication:
1. Novelty of article should be included.
2. In equation (1), why 100% is written? Please correct it.
3. Why do the cohesive index and avalanche angle increase with an increasing rotation speed of the drum?
4. Why the magnesium content in the alloy did not change significantly in the case of both powder and solid samples?
5. Conclusion must be specific. Revise it.
6. Compare the data with published articles in table form.
7. Please check references [4],[16],[17].
8. Please check all references seriously and write in one pattern.
Author Response

(The authors gave the same response as above.)

Reviewer 3 Report
Authors clearly state the problem, a questionable quality of reused materials in additive manufacturing, they examine one alloy in particular, the aluminum alloy powder AlSi7Mg0.6 systematically, and report results in an understandable and well documented way.
The topic is of interest to the readership, the contribution is significant in a way that, for this particular alloy, it debunks the expectations for degraded characteristics of reused materials in additive manufacturing present in the current literature on the subject, and it fits well into the scope of the journal.
Some place for improvement would be in the domain of scientific soundness.
Some minor corrections are related to the text check up:
Acronym definition (line 77)
Sentence correction (line 465)
Typos (line 499)
etc
A more important correction is related to Figure 11. It is too crowded with many lines being very close. I suggest that authors explore different graph layouts like 3D (bars, ribbons...) where the information contained in the graph is more visible.
With all that being said, I suggest the acceptance after minor revision.
Author Response

(The authors gave the same response as above.)

Reviewer 4 Report
Dear authors, I consider that your manuscript needs major revision. Please see the remarks presented in the attached review document.

Author Response
We would like to refer to all of the Reviewer’s comments and the changes made to the manuscript; therefore, all significant changes are highlighted in red colour in the revised manuscript. Please find below our answers to the Reviewer’s specific comments in the attachment.

Round 2
Reviewer 2 Report
The authors have revised the manuscript entitled "Influence of the AlSi7Mg0.6 aluminium alloy powder reuse on the quality and mechanical properties of LPBF samples" very well and have given responses to all questions raised by the referees. Paper may be accepted in its current form.
Reviewer 4 Report
Dear authors, I see major improvements of your manuscript. You answered clearly and satisfactory to all my review remarks. Your research paper can be published in the MDPI journal.